# Research-Based Design Approaches in Historic Garden Renovation

**Albert Fekete [1],* and László Kollányi [2]**

[1]  Department of Garden Art, Faculty of Landscape Architecture and Urbanism, Szent István University, 1118 Budapest, Hungary

[2]  Department of Landscape and Regional Planning, Faculty of Landscape Architecture and Urbanism, Szent István University, 1118 Budapest, Hungary; kollanyi.laszlo@tajk.szie.hu

*  Correspondence: fekete.albert@tajk.szie.hu

**Abstract:** The renewal of historic gardens, landscapes, and sites has grown to be a current issue in Central and Eastern Europe. Based on scientific research, the Department of Garden Art of the Szent István University, Faculty of Landscape Architecture and Urbanism has been dealing with landscape renewal since 1963 on regional, settlement, and garden scales, too. More than 50 years of experience has already proved the advantage of such a research-based design approach in garden and landscape renewal processes, Landscape Architecture has developed from a very practical basis. The purpose of this paper is to show the most significant conclusions of our historic garden research of castle gardens from the Carpathian Basin, focusing on the importance of visual connections designed initially on the sites. Using case studies, the paper intends to explore how proper landscape design in historic environments is achieved. The historical value cannot be simplified or understood as the notion of "old", the heritage being represented by the all-time valuable garden features and elements, independent from their formation in time. In addition to the historical authenticity of the actual use, the social needs and sustainability are important aspects, which must be integrated into heritage protection and reclamation.

**Keywords:** landscape research; view; visual link; castle garden; garden renewal; Carpathian Basin; research and design; historic garden and landscape

---

## 1. Introduction

Rediscovering the scenic value of the landscape has Renaissance roots, and recognizing the aesthetic value of the landscape can be traced in the Renaissance landscape and garden descriptions. The Renaissance image of the world through the conscious observation of nature brings a new chapter in the relationship between man and his environment: he does not only observe the landscape but also perceives its atmosphere and puts his experience of the landscape into words and pictures. Being part of the landscape, the garden belongs to the humanistic approach to life, as a real living space, and as an embodiment of beauty and harmony. The virtual expansion of the garden limits takes place: humanists consciously link the prospect of the landscape into the garden sight, and since then, this has become a consistently applied landscape composition tool—just as in the case of Baroque gardens and landscaped gardens [1–4].

The deliberate shaping of large-scale (landscape) visual connections started already in the Baroque but was accomplished in the Carpathian Basin in the landscaped gardens of the 19th century. The "picturesque" evokes the ideals of the early Enlightenment, a symbolically perceived part of the world. In this period, conscious artistic space organisation resulted in a number of spatial compositions using less functional than picturesque artificial elements (sculptures, buildings, decorations, etc.) in

order to create atmosphere or staffage. These prominent design elements used as a compositional center, which also served as signals and had a symbolic meaning, marked the focal points of the visual axes [5–9].

Regardless of the era, however, it can be stated that during modern garden history, the most important compositional goal of the creators was always the design of a garden image and that of the sight of the garden in time and space; also, the creation of ideal visual links between the garden and the landscape, that is, the view meant to fascinate.

The creation and display of landscape elements is the result of a creative and conscious spatial arrangement. These elements vary depending on whether they are natural or built elements, and from this difference, a different cultural value ensues. Thus, the visual connections ensuring their display are, in some cases, difficult to acknowledge; occasionally, they even transmit different messages to different segments of the society. That is also a reason why cautious and thorough interpretation is needed when assessing their aesthetic value: alongside the examination of the physical appearance, the analysis of the emotions and of the atmosphere evoked by the landscape is also essential [10,11].

The castle garden has thousand-fold connections to its surroundings. At the time of its formation, it was not a simple ornamental garden but featured as an indispensable part of a complex, cultural-historical, ecological and—last but not least—economic system. That is what made it possible for the garden to function in the long run and be sustainable at the same time. Thus, the subject matter of the study is a consciously planned ecological-technical system, both historic and artistic, which can only be interpreted if we embed it into its wider environment. The potential of views and visual axes were taken as features that greatly determined the image of the small area forming the wider environment of the castle park. The statement that "the aristocratic mansion-park ensembles can be considered as a model situation from which the plant and knowledge elements of the model were democratically dispersed on the palace-castle-yard-civic house-farmhouse line" also emphasises the landscape and settlement character of the castle garden, as well as its role in defining the structure [12]. Therefore, when we renew our long-standing gardens while respecting the circumstances of their formation, we need to re-interpret not only the garden itself and its internal relations but their relationship and complex system of connections with the countryside and society as well.

The restoration of the visual axes defining historical compositions is one of the most important tools for the renewal of the historic garden and landscape structure. The paper illustrates and emphasises the essence of restoring the view, and shows that the results of several-decades-long research on the subject can be applied in specific design situations and locations with various features.

## 2. Materials and Methods

The systematic research on Transylvanian castle gardens started in 2004, in cooperation with the National Centre for Historic Monument Conservation and Restoration and has been ongoing research with changing partnerships since then. Until nowadays, the work has comprised the historical research, site survey, analysis, and assessment of 100 Transylvanian castle gardens (Table 1) with the participation of more than 120 university students, many teachers, and external experts contributing significantly to the knowledge base on European garden history [7,13–16].

**Table 1.** The list of castle-gardens investigated during our research.

| Hungarian (Romanian) Name of the Settlement | Name of the Owner Family | Hungarian (Romanian) Name of the Settlement | Name of the Owner Family |
|---|---|---|---|
| 1. Abafája (Apalina, MS) | Huszár castle | 51. Maroskeresztúr (Cristuru M, MS) | Knöpfler castle |
| 2. Alsózsuk (Jucul de Jos, CJ) | Kemény castle | 52. Maroshévíz (Toplita, HR) | Urmánczy castle |
| 3. Alvinc (Vintul de Jos, AB) | Martinuzzi c | 53. Marosillye (Ilia, HD) | Bornemisza c |
| 4. Aranyosgerend (Luncani, CJ) | Kemény castle | 54. Marosnémeti (Mintia, HD) | Gyulay castle |
| 5. Árkos (Arcus, CV) | Szentkereszti c | 55. Marosugra (Ogra, MS) | Haller castle |
| 6. Árokalja (Arcalia, BN) | Bethlen castle | 56. Marosújvár (Ocna Mures, AB) | Teleki castle |
| 7. Bályok (Balc, BH) | Károlyi castle | 57. Marosvécs (Brancovenesti, MS) | Kemény castle |
| 8. Bethlen (Beclean, BN) | Bethlen castle | 58. Marosszentgyörgy (Sangeorgiu de Mures, MS) | Petki-Máriaffy castle |

**Table 1.** *Cont.*

| Hungarian (Romanian) Name of the Settlement | Name of the Owner Family | Hungarian (Romanian) Name of the Settlement | Name of the Owner Family |
|---|---|---|---|
| 9. Bethlenszentmiklós (Sanmiclaus, AB) | Bethlen castle | 59. Marosszentkirály (Sancraiu de Mures, AB) | Bánffy castle |
| 10. Bihardiószeg (Diosig, BH) | Zichy castle | 60. Mácsa (Macea, AR) | Csernovics castle |
| 11. Bonchida (Bontida, CJ) | Bánffy castle | 61. Mezőzáh (Zau de Campie, MS) | Ugron castle |
| 12. Bodola—1 (Budula, CV) | Béldy castle | 62. Mezőörményes (Urmenis, MS) | Rákóczi castle |
| 13. Bodola—2 (Budula, CV) | Mikes castle | 63. Nagyernye (Ernei, MS) | Bálintitt castle |
| 14. Bonyha (Bahnea, MS) | Bethlen castle | 64. Nagykároly (Carei, SM) | Károlyi castle |
| 15. Branyicska (Branisca, HD) | Jósika castle | 65. Nagykend (Chendu, MS) | Schell castle |
| 16. Cege—1 (Taga, CJ) | Wass Á castle | 66. Nagyteremi (Tirimia, MS) | Bethlen castle |
| 17. Cege—2 (Taga, CJ) | Wass J castle | 67. Nyárádszentbenedek (Murgesti, MS) | Toldalagi castle |
| 18. Csákigorbó (Garbau, SJ) | Haller-Jósika c | 68. Őraljaboldogfalva (Santamaria Orlea, HD) | Kendeffy castle |
| 19. Csombord (Ciumbrud, AB) | Kemény castle | 69. Piski (Simeria, HD) | Ocskay-Fáy c |
| 20. Dálnok (Dalnic, CV) | Gaál castle | 70. Pusztakamarás (Camarasu, CJ) | Kemény castle |
| 21. Drág (Dragu, SJ) | Wesselényi c | 71. Radnót (Iernut, MS) | Rákóczi castle |
| 22. Erdőszentgyörgy (Sang de Padure, MS) | Rédey castle | 72. Radnótfája (Iernuteni, MS) | Matskási castle |
| 23. Fiatfalva (Filias,, HR) | Ugron castle | 73. Sarmaság (Sarmasag, SJ) | Kemény castle |
| 24. Fugad (Ciuguzel, AB) | Bánffy castle | 74. Sáromberke (Dumbravioara, MS) | Teleki castle |
| 25. Gernyeszeg (Gornesti, MS) | Teleki castle | 75. Sárpatak (Glodeni, MS) | Teleki castle |
| 26. Görgényszentimre (Gurghiu, MS) | Rákóczi castle | 76. Sepsikőröspatak (Valea Cris, CV) | Kálnoky castle |
| 27. Gyalu (Gilau, CJ) | Barcsay castle | 77. Soborsin (Savarsin, AR) | Nádasdy-Forray c |
| 28. Gyeke (Geaca, CJ) | Béldy castle | 78. Szászfenes (Floresti, CJ) | Mikes castle |
| 29. Gyergyószárhegy (Lazarea, HR) | Lázár castle | 79. Székelyhíd (Sacueni, BH) | Stubenberg castle |
| 30. Gyulafehérvár (Alba Iulia, AB) | Episcopal castle | 80. Székelyszenterzsébet (Elisei, HR) | Kemény castle |
| 31. Hadad—1 (Hodod, SM) | Wesselényi c | 81. Szentbenedek (Manastirea, CJ) | Kornis castle |
| 32. Hadad—2 (Hodod, SM) | Degenfeld c. | 82. Szentdemeter—1 (Dumitreni, MS) | Balási castle |
| 33. Kelementelke—1 (Calimanesti, MS) | Simén castle | 83. Szentdemeter—2 (Dumitreni, MS) | Schell castle |
| 34. Kelementelke—2 (Calimanesti, MS) | Henter castle | 84. Szentgothárd (Sucutard, CJ) | Wass castle |
| 35. Kendilóna (Luna de Jos, CJ) | Teleki castle | 85. Szilágybagos (Boghis, SJ) | Bánffy castle |
| 36. Kerelőszentpál (Sanpaul, MS) | Haller castle | 86. Szilágynagyfalu—1 (Nusfalau, SJ) | Bánffy castle |
| 37. Keresd (Cris, MS) | Bethlen castle | 87. Szilágynagyfalu—2 (Nusfalau, SJ) | Bánffy castle |
| 38. Kerlés (Chirales, BN) | Bethlen castle | 88. Szilágysomlyó (Simleu Silvan, SJ) | Báthory castle |
| 39. Kisbún (Boiu, MS) | Bethlen castle | 89. Székelyudvarhely (Sambatesti—Odorheiu Secuiesc, HR) | Ugron castle |
| 40. Koltó (Coltau, MM) | Teleki castle | 90. Szurdok (Surduc, SJ) | Jósika castle |
| 41. Koronka (Corunca, MS) | Toldalagi c | 91. Tasnád (Tasnad, SJ) | Cserei castle |
| 42. Korpád (Corpadea, CJ) | Gaál castle | 92. Torockószentgyörgy (Trascau, AB) | Thoroczkay- castle |
| 43. Kővárhosszúfalu (Satulung, MM) | Teleki castle | 93. Vajdaszentivány (Voivodeni, MS) | Zichy-Horváth c |
| 44. Kraszna (Crasna, SJ) | Cserei castle | 94. Válaszút (Rascruci, CJ) | Bánffy castle |
| 45. Kutyfalva (Cuci, MS) | Degenfeld c | 95. Váralmás (Almasu, SJ) | Csáky castle |
| 46. Magyarbükkös (Bichis, MS) | Kemény castle | 96. Várfalva (Moldovenesti, CJ) | Jósika castle |
| 47. Magyarcsesztve (Cisteiu de Mures, AB) | Mikes castle | 97. Zabola (Zabala, CV) | Mikes castle |
| 48. Magyarfenes (Vlaha, CJ) | Jósika castle | 98. Zám (Zam, HD) | Nopcsa castle. |
| 49. Magyarózd (Ozd, MS) | Radák-Pekry c | 99. Zsibó—1 (Jibou, SJ) | Wesselényi castle |
| 50. Magyarpécska (Pecica, AR) | Klebelsberg c | 100. Zsibó—2 (Jibou, SJ) | Béldy castle |

## 2.1. Work Methodology

The survey methodology of Transylvanian castle gardens was based on the principle that the sites concerned must be interpreted in context with the related settlements and landscapes, as the only way to understand their historical importance and current value.

For a systematic survey of the visual links and eye-catchers of the castle gardens, we established the following theoretical framework:

- Identification of all potential gardens,
- Definition of the priority list of the gardens,
- Historical research of the gardens,
- General landscape assessment of the present conditions of the gardens and its environment,
- Survey and assessment of the nowadays area, the spatial layout, the composition, the visual links of the gardens and its surroundings.

The research methodology may be practically divided into two main parts: the garden history research and the site survey.

### 2.1.1. Garden History Research

The goal of the historical research of primary and secondary sources found (archives, library and museum materials, map and postcard collections, thematic bibliography reviews, internet sources, etc.) is to provide a clear idea of the establishment and development of the gardens. It comprises the role the sites play in the landscape and the urban character and layout, and the landscape scale relationships that served as a basis for the establishment of the castle garden and determined the character of the surrounding landscapes to a great extent. The garden history research also deals with the architectural history of the castle and the family history of the owners. Family history data proved to be especially important, as it was tightly linked to the initial creation or later transformation of parks, certain garden sections, or specific garden features. The owner families are the bearers of the intellectual and cultural substance that is essential for the spirit and identity of the place, and for the establishment of the castle gardens. Many of the gardens were shaped from the ideas of the owners, or the design was directly influenced by the owners. Owners of the castles and gardens may therefore also be considered as creators of these historic monuments. The results of the garden history research are the inventory and the cultural heritage assessment.

### 2.1.2. Site Survey

The site survey precisely records the actual conditions of each castle garden (sketches, minutes, GPS coordinates, geodetic surveys, geophysical surveys, aerial remote sensing survey, plant inventory, digital photographic inventory, etc.) as well as the valuable features, and thus, it serves as a status report or basis for comparison for conservation strategies and any future restorations. A topographic map (land registry map, etc.) provides the basis for the survey of the general conditions and valuable landscape features. Definition and classification of the survey criteria were important steps of the procedure. As a starting point, we took the criteria of historical heritage surveys in Hungary, complementing and adapting it to local circumstances as necessary. The survey form records aspects of heritage conservation, visual landscape protection, and dendrology, focusing primarily on the valuable botanical, architectural and landscape features, visual links, and spatial composition. The prepared survey form also integrates the recommendations from the English Heritage and the National Centre for Historic Monument Conservation in Hungary.

In the case of one of the most famous and influential British gardens, the Rousham House Garden, we can also bring up the eye-catchers and defined visual axes as the most significant compositional tools—as proof of their success in garden history to this day: "The many wandering walks through the gardens are full of delicious surprises, a sudden meeting with a dying gladiator, a glimpse of Apollo, or a long view of a Gothic mill, an ancient bridge or distant trees, or arrival at an unexpected seat in an alcove", says Hal Moggridge, an English landscape architect [17].

Concerning Transylvanian castle gardens and landscapes, we tried to determine those prospects, eye-catchers, and visual axes in cases of 100 locations, which through their meanings and symbolic messages play an essential role in the garden composition or landscape they are part of.

Similarly to Hungarian and other examples from the countries in Europe [18], planned visual links in Transylvania are specific mostly to landscape gardens from the 19th century, a period that was also the golden age for Transylvanian castle gardens (Figure 1).

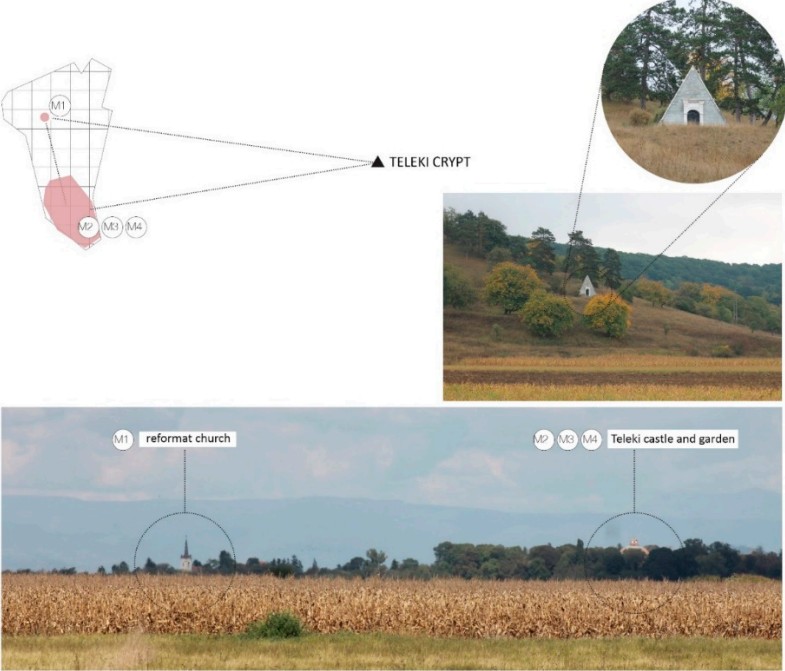

**Figure 1.** Visual attractions represented by built features in the case of the Teleki estate from Gernyeszeg (Gornesti, R.O.). Photos and graphical processing by Fekete, A., based on [19].

The deliberate spatial layout resulted in landscape compositions with structures and ornamental elements that were not really functional in themselves; instead, picturesque elements were applied as eye-catchers, staffage, and effects to create a specific atmosphere.

As a highlight of the composition, the eye-catchers also played an important role in the case of the designed landscapes in Transylvania, providing the focus for visual axes, occasionally having an additional, symbolic meaning.

## 3. Results

Besides being aware of garden history, the expert needs a thorough knowledge of the current situation, a regard for the value that still exists, as well as detailed on-site research and survey in order to deal with garden renewal.

Recognition and interpretation of historical value are also dependent on environmental factors. Relatively intact gardens that are easy to research and document pose a straightforward resolution case for the designer. On the other hand, when territorial integrity has been impaired or the historic value transformed (built-in, overgrown, dilapidated, etc.) and fallen into ruins, the authentic renewal is a complex and responsible task since the monument conservation must properly be carried out alongside missing data supplementation or completion. Reconstruction of a garden or garden element can only occur if no other memory needs to be destroyed for its sake, and there are sufficient genuine pieces of information available. The Florence Charter mentions this: 'In principle, no one period should be given precedence over any other, except in exceptional cases where the degree of damage or destruction affecting certain parts of a garden may be such that it is decided to reconstruct it on the basis of the traces that survive or of unimpeachable documentary evidence" [20].

Landscape architecture is an applied science. According to the guidelines on the preparation of scientific publications in garden history, the results of the research are based on general preliminary studies of garden and landscape history, the research results and experiences of several decades, the exploration and analysis of authentic historical sources, and the site surveys and assessments.

*3.1. Thesis No 1: The Castle, the Castle Garden, and the Surrounding Landscape Altogether Represent a Single Artistic and Compositional Unit*

From the 100 investigated locations in 79 cases, during the field survey, we discovered strong visual connections still existing between the castle, the garden, and the landscape.

Neither the castle or the castle garden should be interpreted independently. They are a single unit, and all the man-made and natural elements of the castle garden—often specific elements beyond the garden boundaries—make part of this unit. The castle and the garden altogether represent a composition that is an integral part of a complex system developed on artistic, cultural, historical, ecological, and economic bases. When the estate complexes were established, the views and visual axes made use of the landscape potential, relying on architectural and artistic tools to explore.

*3.2. Thesis No 2: Visual Links Applied to Castle Gardens at a Landscape Scale may Be Divided into Two Main Categories: Eye-Catchers and Prospects*

The views were produced by creating effects with the aid of visual tools, and by delivering symbolic meanings and messages. Regarding the planned visual links between the landscape and the garden, two types are possible to distinguish: the "eye-catchers" and the "prospects".

The eye-catchers are structures that draw attraction even from a greater distance and are applied as focal points of a visual axis or determine a specific visual link. The eye-catchers—as distinctive elements of the landscape—determine the structural layout of landscape gardens. In the periods of the 19th-century Sentimentalism and Romanticism, characteristic structures and buildings in the garden were used not only for functional but also for spiritual, political, and aesthetic reasons [21–23].

The definition of a prospect as a picturesque scene resulted from the large scale composition of designed landscapes comprising the skyline of the natural or built environment, the vegetation, and the water surfaces [9].

The prospect is the virtual extension of the garden boundaries, the inclusion of the surrounding landscape into the garden scenery. Very often, the same landscape composition allows for the definition of several representative visual axes and several connections [10]. This is especially true for larger garden landscapes—at this point, we can refer to an excellent foreign example, the most extended European garden landscape, the English garden of Dessau-Wörlitz, where Edith Kresta mentions more than 300 visual axes applied as parts of the composition [24].

Out of 100 investigated locations in 61 cases, altogether, we found 139 built features (eye-catchers) on site, which proves the garden and landscape compositional role of eye-catchers and can serve as a basis for the renewal of the visual communication between garden and landscape. (Figure 2).

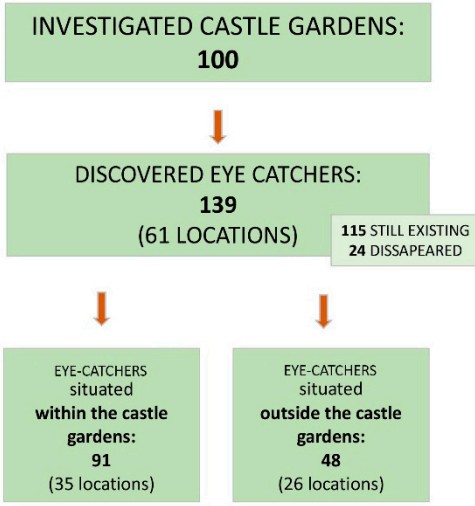

**Figure 2.** Eye-catchers location in Transylvanian castle gardens.

In several cases, we discovered former eye-catchers as well, which unfortunately disappeared in the course of the last century (Figure 3).

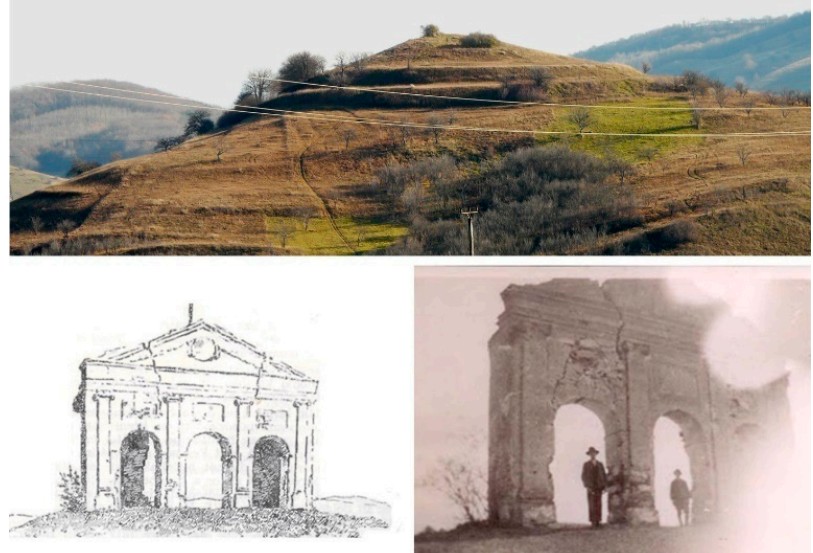

**Figure 3.** The Csicsal Hill in Magyarfenes (Vlaha) nowadays (photo by authors, 2018), without the Gloriette of the Jósika Castle Garden, a vanished eye-catcher represented by a sketch (bottom left) based on a photo (bottom right) [25].

### 3.3. Thesis No 3: Eye-Catchers Are Predominantly Architectural Structures

Eye-catchers applied as terminal points of planned visual axes of castle gardens are architectural or artistic works that draw attention even from a greater distance. They can be identified and classified (Figure 4).

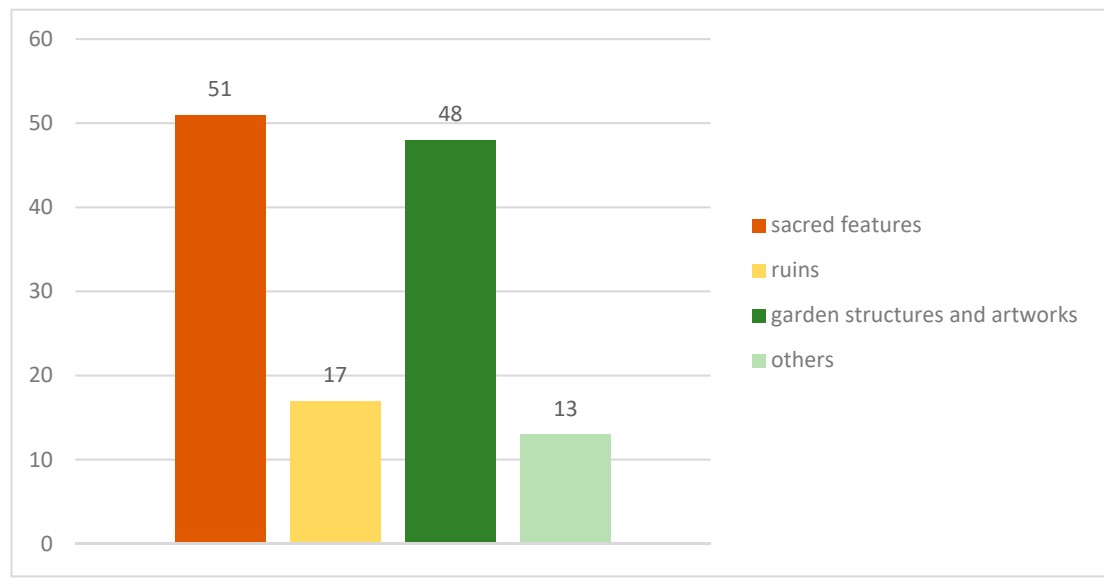

**Figure 4.** Graphical representation of the main eye-catcher types defined during the Transylvanian Castel Garden Research and its numbers.

The eye-catchers applied most frequently in the castle gardens are:

- Sacred features/eye-catchers (church towers, burial monuments, tombs, memorial places, crypts, gloriettes, tempiettos, obelisks, etc.), altogether 51 (Figure 5);

- Ruins (of fortresses, castles, manor houses, churches, etc.), altogether 17;
- Garden structures and artworks (gazebos, pavilions, fountains, ornamental pools, cascades, grottos, garden cottages, flights of stairs, balustrades, sculptures, viaducts or other structures of staffage, and built garden features), altogether 48;
- Others, altogether 13.

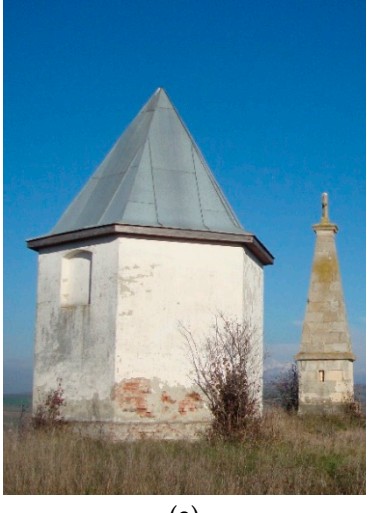 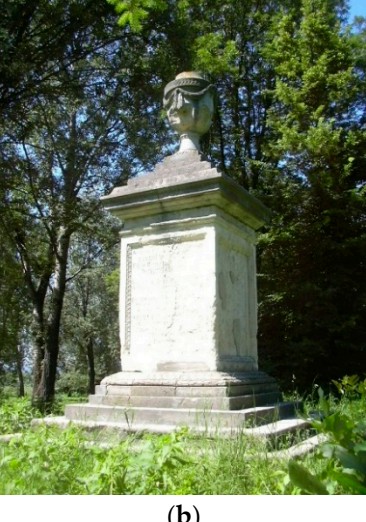 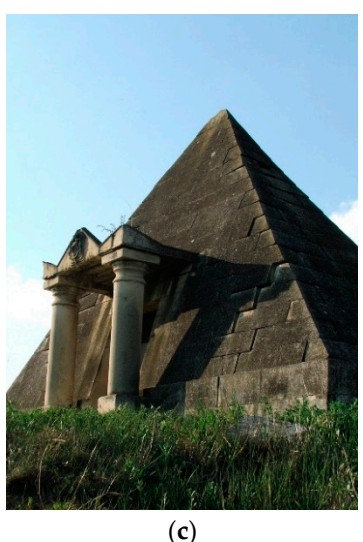

(**a**)                (**b**)                (**c**)

**Figure 5.** Sacred eye-catchers: Bethlen Family Crypt, Nagyteremi (Tirimia, RO) (**a**); Memorial urn of József Teleki, Gernyeszeg (Gornesti, RO) (**b**); Pyramidal tomb of the Kemény Family, Csombord (Ciumbrud, RO) (**c**); Photos by Fekete, A, 2013 (**a**), 2016 (**b**), 2017 (**c**).

*3.4. Thesis No 4: Due to the Scarce Sources in Garden History, Garden Reconstruction Is Rarely a Feasible Option in Carpathian Basin. Other Types of Research-Based Design Approaches Have to Be Defined in Order to Ensure the Authentic and Value Preservation Oriented Historic Garden Renewals*

On the basis of the information available on the garden, the relevant heritage conservation principles, and the user needs, we have distinguished the following garden reconstruction categories for historic gardens:

- Garden renovation (revitalisation),
- Garden regeneration,
- Garden restoration.

## 4. Discussion

Results of the research carried out through several decades on historic gardens and landscapes can and ought to be applied to specific planning situations. That is why this chapter comprises selected examples[1] of projects where the historic value of the planned visual link between the garden and its surroundings is possible to identify, the compositional value of the visual axes and prospects is high and increases the aesthetic quality of the manor garden and the surrounding landscape; also, the visual features bear some symbolic meanings.

The projects for various sites introduce some specific historic garden reconstructions, which demonstrate the potential planning approaches required in a given situation for all garden reconstruction categories established. Through the examples, I intend to systematically summarise and evaluate the

---

[1] The authors plans realised.

garden reconstruction theories in Hungary at the beginning of the 21st century and to formulate new findings and proposals on the contemporary principles on historic garden restoration and planning.

The introductions of the renewal projects are not comprehensive: partly due to the limitations on length, and partly to focus on the topics (the garden scenery and the planned views) of the paper, only the relevant details of the projects have been included. Thus functional, technical and infrastructural, (roads, pavements, public utilities, lighting, earthworks), ecological and dendrological (plant geography, dendrology, plant associations, plant species applied, etc.), pedological, hydrological, hard landscaping, above ground drainage, and design aspects are introduced only to an extent necessary to have a general understanding of the specific example.

### 4.1. Garden Renovation (Revitalisation)

Garden reconstruction is based on the historical sources available, the heritage features of the site that are possible to identify, and the stylistic elements and analogies of a specific period. The park is renovated with a distinct application of the stylistic elements of the period most relevant to the site and with additional functions to meet the actual demands.

A good example of the renovation of the internal and external axial views is the Mikes Castle Garden in Zabola (Zabala, RO) [26].

The Mikes Castle in Zabola was built by Zsigmond Mikes in the 1620s as a two-storey Renaissance building with a quadratic layout. In the course of time, the castle underwent several renovations. Its current appearance dates back to 1867 when Benedek Mikes renovated the old building in Neoclassical style and extended it into three storeys. Both the castle and the 36 ha dendrological park that surrounds the building that were established in the 1880s in late landscape garden style are listed heritage assets. The property was nationalised in 1949, and both the buildings and the park significantly decayed in the second half of the 20th century. Elements of the listed heritage were irrecoverably damaged. The 2002 restitution returned the residence to the rightful owner, opening the opportunity for the renovation of the castle and the park.

In the course of the garden history research, it was possible to discover sources that could provide a basis for the reconstruction of the original spatial structure that had been only partially preserved by the beginning of the 21st century.

Two plans served as that foundation for the renovation of visual links. The construction of the park was started by the owner, Count Benedek Mikes in the 19th century, thereafter continued and completed by his son Ármin Mikes. A garden masterplan scaled 1:1000 and named *Blatt zur Konzeption II.* has been preserved from this period (Figure 6a). Although some of the details have never been realised, other details justify the appropriateness of the design concept, even nowadays. The title of the plan also hints at there having been a first development concept (presumably named *Blatt zur Konzepzion I.)*, which has not been found. A key person of the creation of the castle garden was Achille Duchene[2], a renowned French landscape architect [27].

Matched against the historic photographs, the old plans revealed the full layout of the terraces, which provided ground for the then functional (tennis courts, leisure grounds) and ornamental (rose garden, carpet beds) units, and also organised the visual links (Figures 6b and 7).

---

[2]  Achille Duchene (1866–1947), was a master of French geometric gardens in the first decades of the 20th century. His works may be found worldwide. He designed formal landscape gardens with distant and framed views. Main sites of his projects include: The Wigwam Garden, Villa Ephrussi de Rotschild, The Carolands, Courances Garden, Vaux le Vicomte Garden, Champs sur Marne, Courances and others. Also, he was the one who made the restoration plan for the water parterres of Blenheim Palace in England.

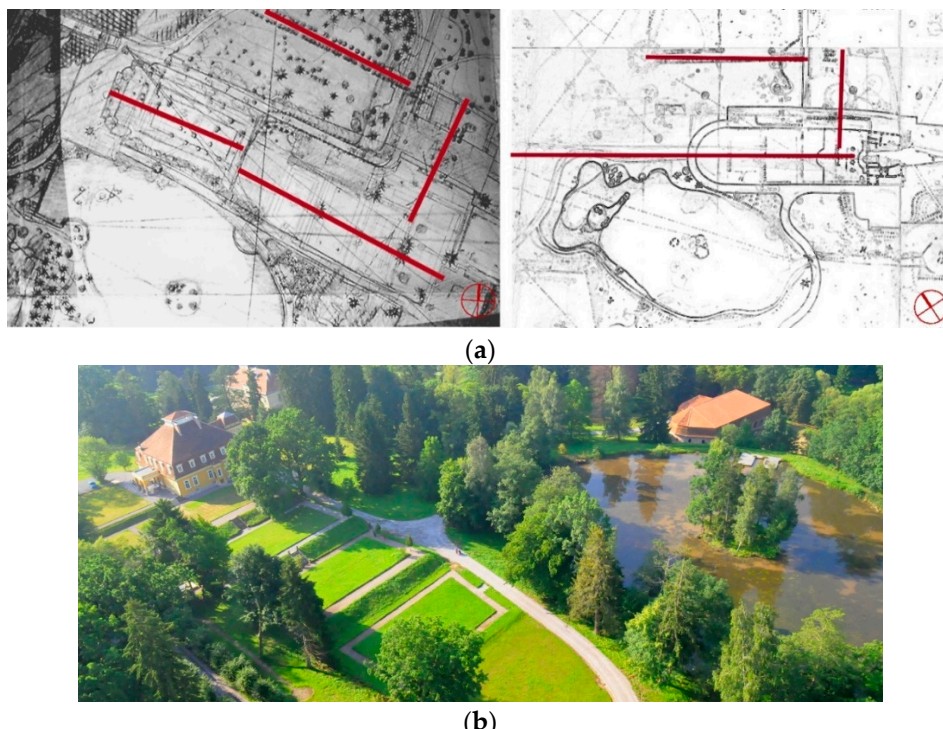

(**a**)

(**b**)

**Figure 6.** Renovation of terraces (**b**) and visual connections in the case of the Mikes Castle Garden in Zabola (Zabala, RO), based on analysis of old plans (**a**) and field survey. Design: Fekete, A.; Magdó, J.; Jávori, K. 2008–2010. Photo by the Mikes Family, 2014.

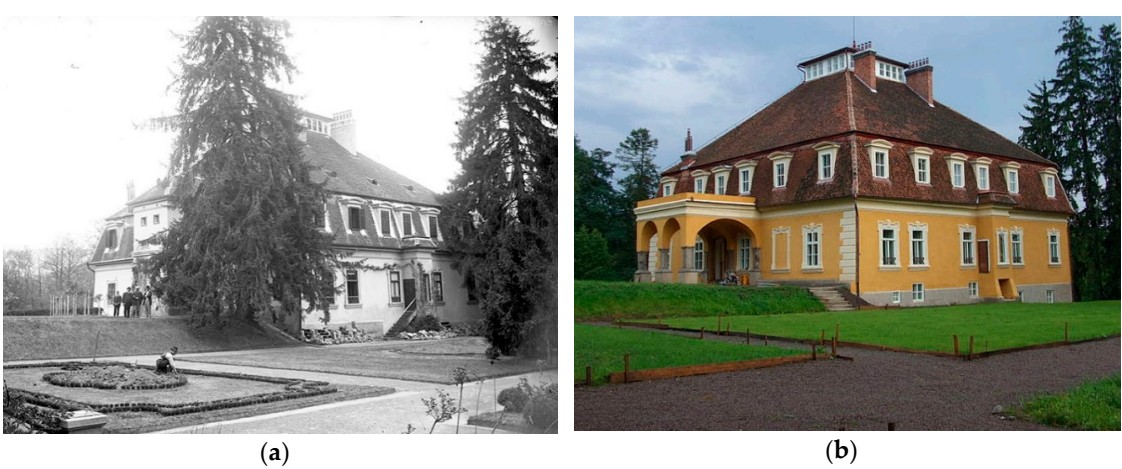

(**a**) (**b**)

**Figure 7.** Terrace renewal in a minimalistic style, Mikes Castle Garden, Zabola (Zabala, R.O.). Source: Photo Mikes Family Archive, 1911 (**a**); photo by Fekete, A., 2013 (**b**).

With reference to the original plans preserved, an authentic renovation plan was prepared, with emphasis on the reconstruction of internal and external visual links. The geometric system of the terraces has been completely reconstructed, with the original mass and visual links restored. Nevertheless, taking into consideration new user demands, the terraces have been renovated without restoring also the former ornamental elements (rose garden, carpet beds, trimmed evergreen, and deciduous hedges) of the garden, with only lawn-covered horizontal surfaces and slopes, using and renovating the old structures of stairs (monolith stairs and columns, balustrades) through a minimalistic garden regeneration. The heritage assessment has proved that the site fully meets the criteria for historic gardens; therefore, as necessary in such circumstances, principles of heritage conservation

must be applied. The decision for garden regeneration as the most appropriate solution is supported by the following facts:

- Since the 2002 restitution of the manor, the original residential function is only partial (old castle);
- For long term economic sustainability, the estate must maintain an important new function (as tourist accommodation): several renovated buildings of the estate (former boiler room, stable, shed, powerhouse, Swiss house) and the new castle all provide touristic services;
- The terms of renovation grants have necessitated the provision of additional new functions (venue, cultural centre);
- Uses arising from the new functions result in increased seasonal and occasional (events) loads that the garden should be capable of coping with;
- Residential functions and hospitality services should also be spatially separated in the castle garden, which requires an appropriate structural and functional layout of the garden.

In addition to serving the new functions, the main objectives of the design programme established in cooperation with the client were to restore the authentic spatial layout, atmosphere, and visual links that were specific to late landscape gardens, and also to renovate the related water infrastructure and the geometric system of terraces from the first years of the 20th century, adjacent to the old castle.

In the first half of the 20th century, the castle garden included a complex of wetland habitats (several ponds linked by watercourses and various structures used for water management). As a result of inappropriate management, most of the water surface had disappeared by the second half of the century, and a shrubby, woody vegetation took over the area of the former ponds. The renovation of the pond system and the wetland habitats has not only reinstated the ecological equilibrium and biodiversity but has also regenerated the internal visual links of the manor garden across the waters. Multiple versions of plans were prepared for the reconstruction of the water infrastructure (Figure 8).

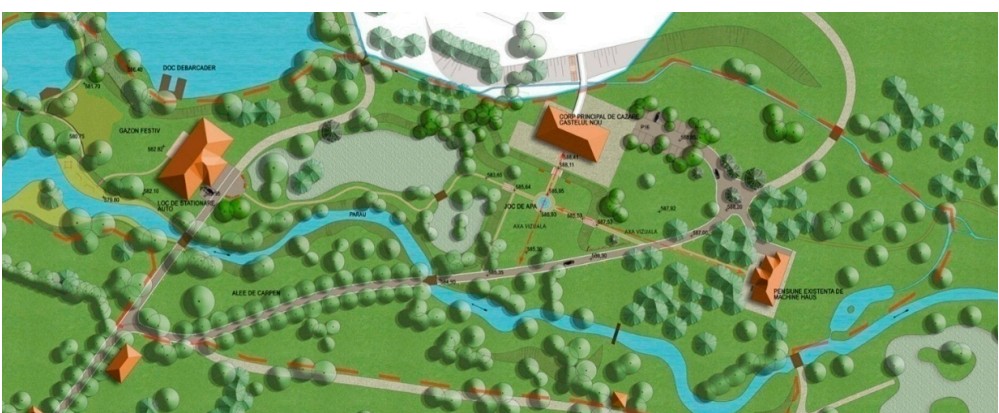

**Figure 8.** The final proposal for the revitalisation of the water infrastructure (detail), with the phases scheduled, Mikes Castle Garden, Zabola (Zabala, R.O.); Phase 1—blue; Phase 2—grey; design: Fekete, A.; Magdó, J.; Jávori, K. 2008–2010.

The overgrown shrubby vegetation and the water system has been cleared throughout the garden to a necessary extent, leading to the renovation of the spatial composition (Figure 9). The careful management of the woody plants has opened the space occupied by invasive plants for the dendrologically valuable tree specimens and also started the restoration of the generous views characteristic for landscape gardens. Long bygone visual axes and views important for spatial layout and composition have been revived, the connection between the garden and the landscape is seamless, boundaries dissolve, and views are undisturbed.

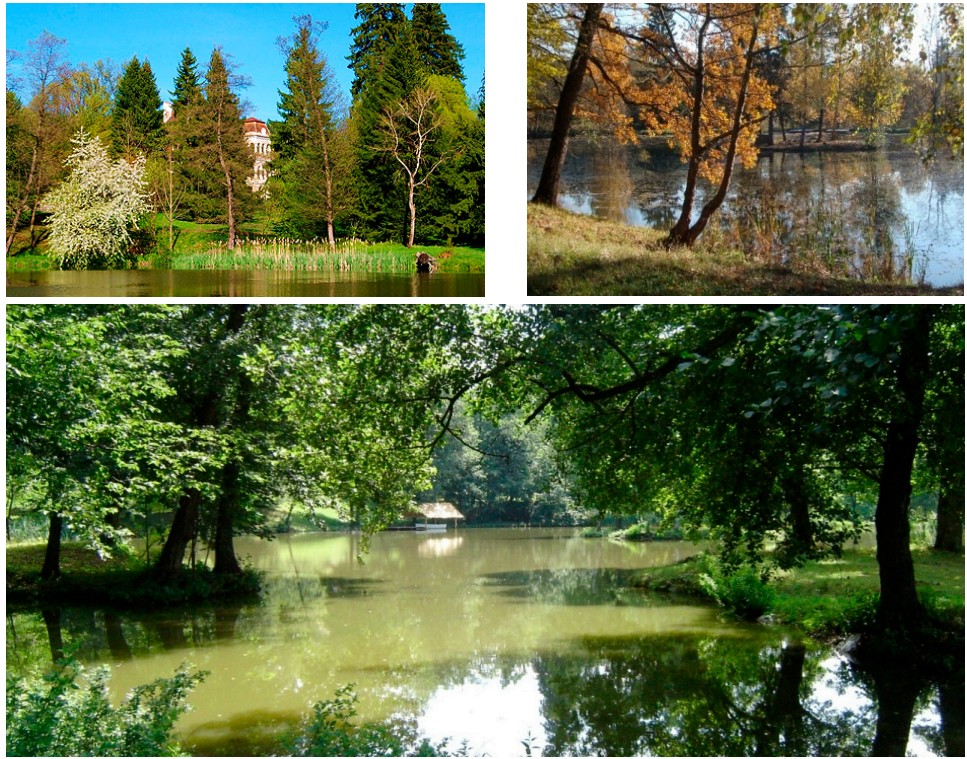

**Figure 9.** Renewed pondside views and visual axes through the water mirror in the Zabola Castle Garden. Photos by Fekete, A., 2018.

*4.2. Garden Regeneration*

The park/garden that has lost its historic features to a great extent is reconstructed on the basis of historical sources and analogies available. The solutions were applied to integrate the features of the relevant historical period into contemporary contexts. This approach establishes the opportunity of creating a contemporary work of art.

A good example of the historic garden regeneration is the case of the Regeneration of the Herb Garden and Arboretum of the Pannonhalma Archabbey, Hungary [28].

The project aimed at the regeneration of the 10-hectare Lavender Garden, and the related Herb Garden to the east of the Arboretum was also established around this time.

Similarly to the historical traditions of other orders, the Herb Garden always had an important role in Pannonhalma. Therefore, the introduction of the long traditions of the Pannonhalma Benedictines in medication and herb cultivation was a priority of the development, which has been complemented with the production of lavender oil in the past decades.

Our landscape design around the horticultural buildings of the archabbey recalls the atmosphere of traditional farmyards. The courtyard was fenced around by low stone walls, with a distinct separation of the outer versus inner areas, the farm buildings from the cultivated land. The show garden of herbs also reflects this approach. The strict geometry of the layout, the allocation of the planting beds, and the static water surface of the garden pool are all applied to strengthen the feeling of functionalism, while its scales recall the atmosphere of the historic monastery gardens. Nonetheless, with respect to the use of materials and the design of garden structures (pergola, ornamental pool), we applied contemporary solutions that are also capable of coping with the tourism load. Authentic depictions of the herb garden regarding its extent, exact location, and former design, which could have served for a direct restoration, were not available. That is why the garden was regenerated by analogies.

The layout, structures, and furniture of the garden are all geometric: benches are block-like, while the hard surfacing applies the same minor cobble stones and gravel that are general throughout the abbey. Retaining walls are all made with the grey limestone cover.

In the course of the planning, the reinforcement and highlighting of the relevant internal and external eye-catchers (the abbey tower, the new building of the distillery) and the views (towards the Arboretum and the surrounding landscape) had a priority. Faithfully to historical traditions, with the application of specific landscape design tools and accents in the composition, certain sections of the garden (e.g., the Herb Garden) were related distinctly to the abbey, while others (lavender fields) to the surrounding landscape (Figure 10).

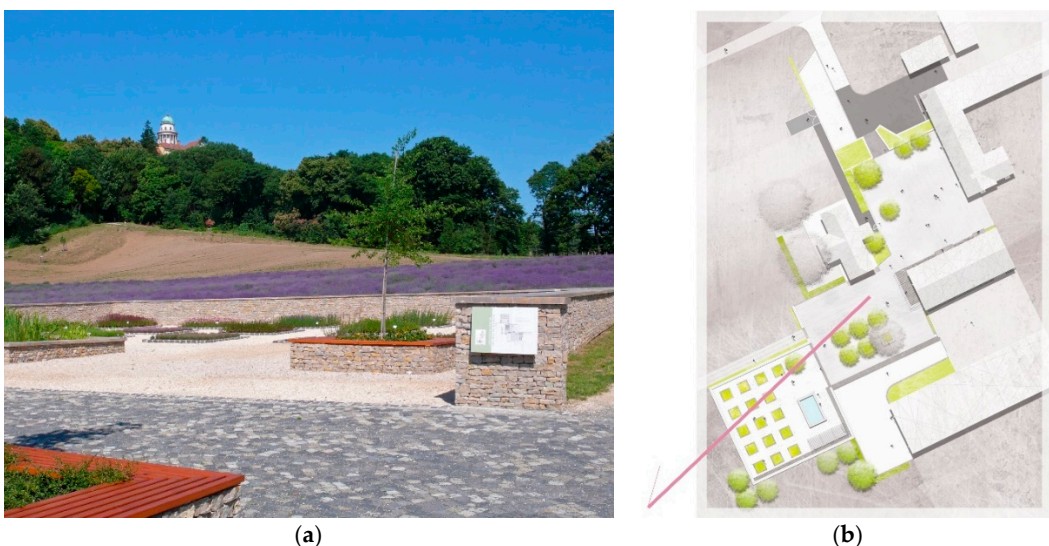

(**a**)　　　　　　　　　　　　　　　　　　　　　(**b**)

**Figure 10.** Garden regeneration in the case of the herb garden in Pannonhalma Archabbey (HU). The view of the Cathedral's tower in the background suggest the sacred character of the site (**a**). The sub-Figure 11b shows the masterplan of the herb garden, marking with red line the renewed view towards the Cathedral's tower. Design: Fekete, A.; Vajda, Sz.; Szilágyi, K. 2010–2011. Photo by Fekete, A., 2014.

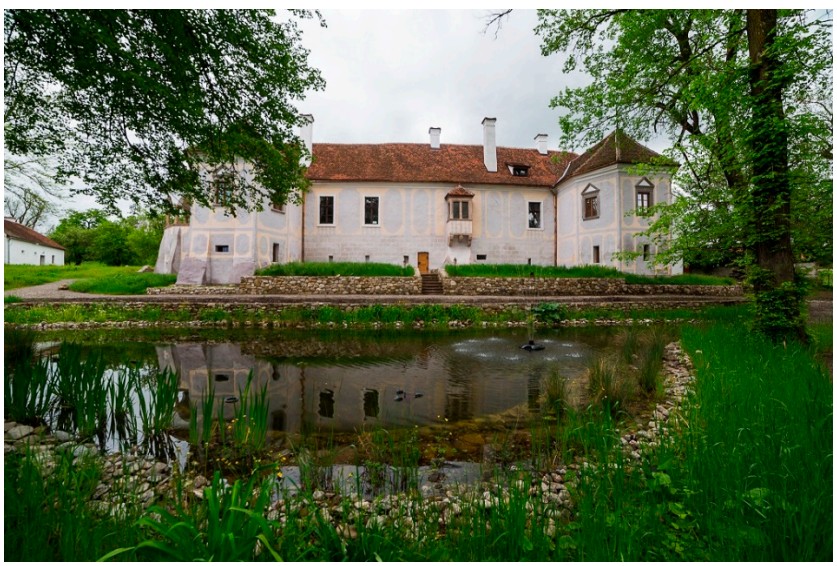

**Figure 11.** The restored pond, with the manor house in the background. Design: Fekete, A.; Rudd, M., Sárospataki, M., Weiszer, Á.; 2015. Photo by Fekete, A., 2017.

A second example of garden regeneration is the Kálnoky Castle Garden reconstruction in Miklósvár (Miclosoara, R.O.) [29]. The castle was built in Renaissance style in the 17th century. Despite several later Classical renovations, the building has preserved its Late Renaissance character. Owing to its structure and decoration, it is a noticeable monument in the Háromszék Region, reflecting representative functions of manor houses of Secler noblemen in the 17th and 18th centuries. The building is listed, managed by the former owner Kálnoky family.

Based on an analysis from the garden historical assessment criteria introduced in the doctoral thesis, the 14 ha manor park is a garden of historical value, and, according to descriptions and registers from the end of the 17th (1698) and beginning of the 18th (1716) centuries, its northern part adjacent to the manor house was a Renaissance garden with typical garden structures of the age. We have found clear descriptions or references to the pond, the gazebo on the peninsula, the wooden bridge, and the parterre flower garden [30].

The renovation of the garden was based on the following facts and findings:

- After the renovation, the use of the manor house and garden will be different from the original residential function;
- In the future, the function of the manor will be representative: partly as a cultural centre (a museum of the Transylvanian villages) and partly as a hotel;
- Uses arising from the new functions result in increased seasonal and occasional (events) loads, which the garden should be capable of coping with; the garden, as the first renewed Late Renaissance manor garden in Transylvania, should authentically recall the atmosphere of the 17th century Late Renaissance and serve as a good example for similar subsequent renovations.

In addition to serving the new functions, a main objective of the design programme established in cooperation with the client was to restore the functional units, layout, and axial views that were specific to Late Renaissance Transylvanian gardens. In this context, the renewal of the pond and the related garden structures (gazebo, wooden bridge) and the parterre garden was also proposed.

Based on historic records discovered during the garden history research, original functional units, layout, and axial views dating back to the 17th century were possible to restore so that the new functions were also properly supplied.

For the restoration of the pond and the retaining walls, beyond the evidence of written records, landform analysis has also confirmed the location of the pond during the site survey. Fed by a creek, the pond was located at the lowest part of the site, close to the creek.

As an open surface with visual links, the pond is an important element of the spatial structure of the garden, providing emphasis to the built features and supporting planned views. Its reflective surface also heightens the visual effects and experience (Figure 11). At the same time, the pond also served for economic purposes and used as a fish pond as it was typical for ponds in manor gardens in this period.

The castle is located at an elevation of 2.5 m above the pond behind the house. This position is beneficial not only for flood prevention and static reasons but also for the appearance and garden views: providing an accent to the castle that is, therefore, visible also from beyond the garden.

The levels of the castle and the pond were connected with a double line of parallel retaining walls. In the course of time, similarly to the dried bottom of the pond, the retaining walls also became filled and covered. Their line was, however, possible to be traced by the landform. The original location and structure of the retaining walls and the pond bottom were identified through archaeological survey, using an archaeological trench. The reconstruction of the retaining walls was based on the excavation and study of the structures well preserved underground. (Figure 12).

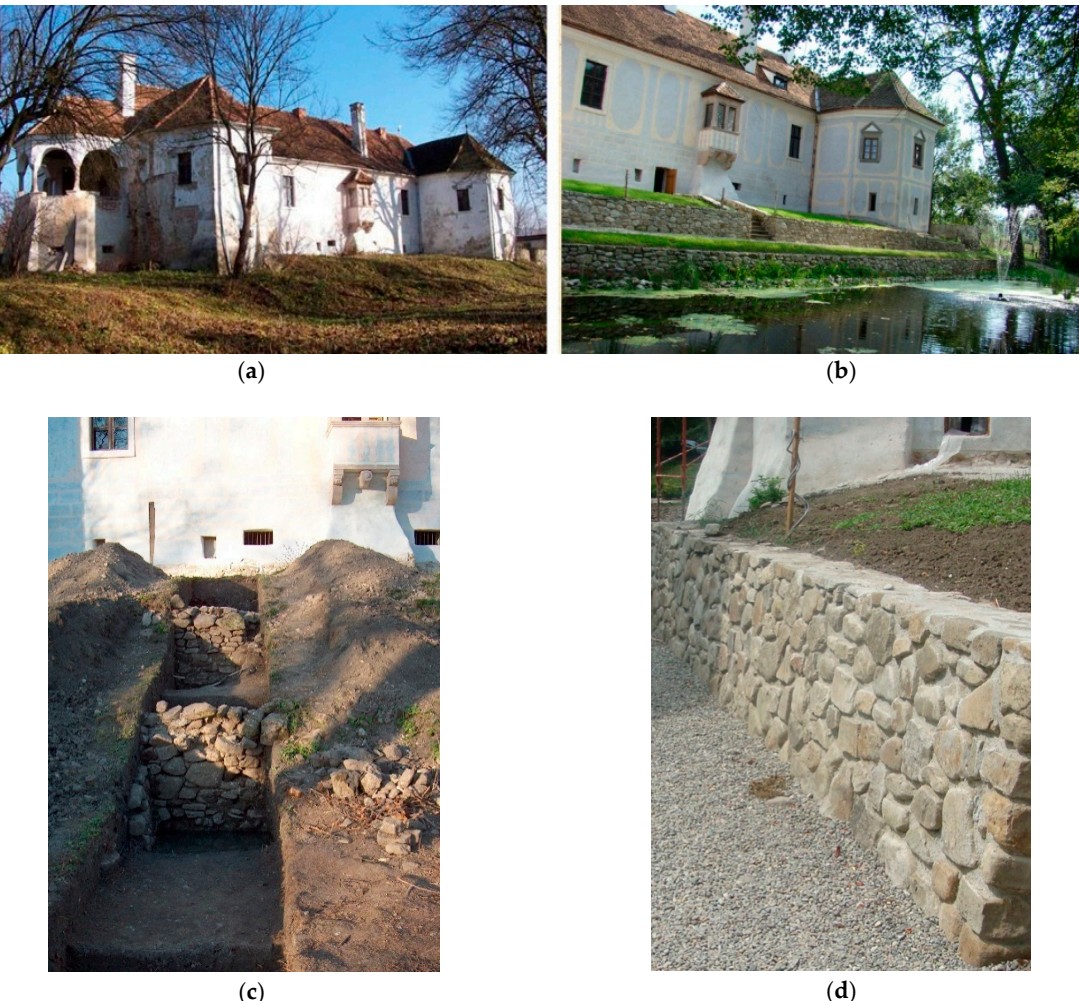

**Figure 12.** The castle facade towards the pond with the retaining walls in 2014 (**a**) and after the reconstruction in 2016 (**b**). The archaeological excavation of the retaining wall (**c**) and the renewed retaining wall (**d**). Photos by Fekete, A, 2017.

The gazebo is a pavilion-like roofed garden structure, mentioned in the 16th and 17th-century garden descriptions from Northern Hungary, Southern Hungary, and Transylvania [31].

The gazebo as an eye-catcher was also a basic feature in the late renaissance garden. The regeneration design of it was based on descriptions of gazebos from other Late Renaissance gardens in both Transylvania and the historical Upper Hungary, and also on the study of the characteristic elements of the 17th-century Transylvanian architecture, the wooden belfries that have been preserved at various places.

Despite the fact that written sources of the octagonal gazebo are scarce, it was possible to reconstruct it so that the size, shape, structural details, and materials are authentic (Figure 13).

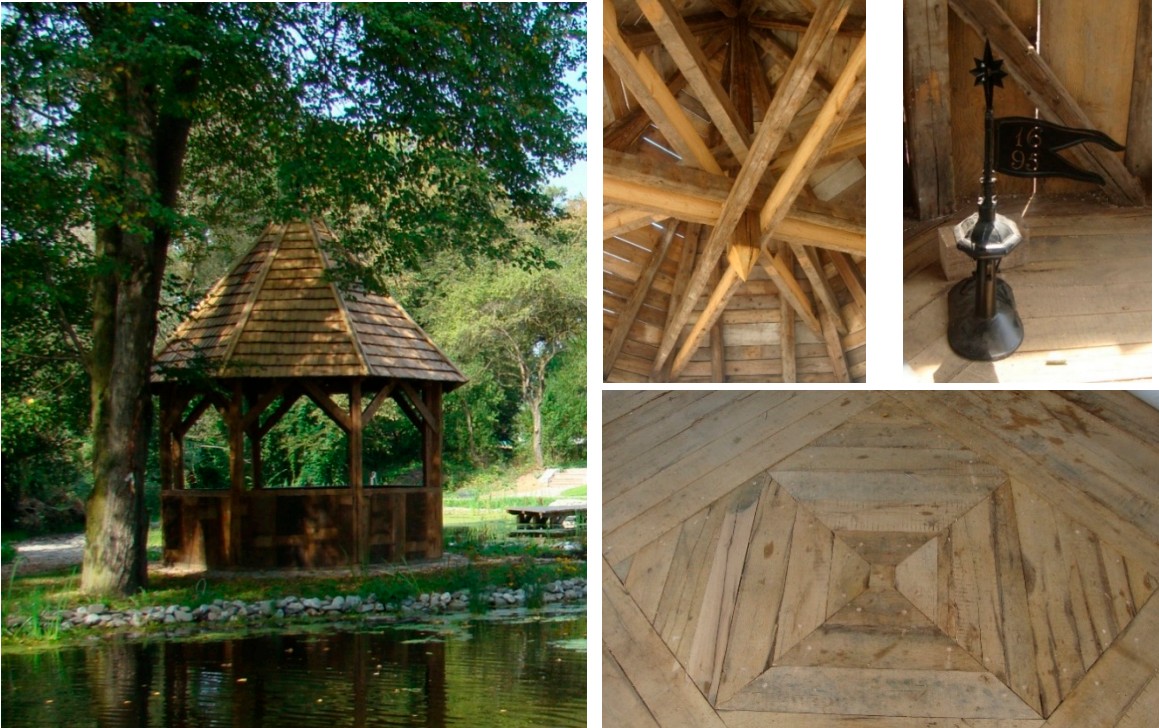

**Figure 13.** View and technical details of the gazebo in Miklósvár, constructed according to historical descriptions and analogical examples and used as an eye-catcher. Photo by Fekete, A., 2017.

*4.3. Garden Restoration*

Garden reconstruction was based on the exploration of historical sources, the archaeological surveys, the use of preserved garden features, etc. Based on the preserved historic features and the available historical sources, it is possible to restore a part or the whole of the garden so that it will be greatly identical to the original design.

A good example in this category is represented by the restoration of the Upper Garden of the Royal Palace in Gödöllő, Hungary, which is one of the most valuable gems of Baroque in Hungary [32]. Construction of the Palace at Gödöllő was launched in 1735, under the direction of András Mayerhoffer. A unique feature of the Palace is the "inside out" nature of the layout, as the cour d'honneur does not serve as a reception area but looks towards the garden. Embracing the representative courtyard, the U-shaped building of the Palace provides a concave shape from the direction of the garden, which is ideal for a Baroque spatial composition.

The restoration strategy—following on from the above, and also with regard to the current and future functions of the garden—aimed at the restoration of the spatial composition and atmosphere that served for representation and recreation in the 19th century. Although the landscape garden has been reconstructed several times during its nearly 200-year history, the spatial layout and composition of the picturesque garden from the beginning of the 19th century has been preserved to a significant extent. Just as in the picturesque garden some significant features from the earlier Baroque have been retained, such as the pavilion on the Royal Mound, the horse chestnut alley, and the Shooting Gallery. In the last third of the century, promoted to a royal residence, Gödöllő became an exemplar of cultural and artistic representation of the age throughout the country. In the Romantic landscape garden established during this time, the naturalistic spatial layout and composition has been kept, while the use of plants was also moderate compared to the often rampant diversity of collector's gardens.

Regarding the planning, it was, therefore, necessary to go beyond the usual restoration clichés and concepts with an overwhelming priority of functional aspects. An artistic renovation based on the historical plant material and composition principles was the appropriate solution (Figure 14).

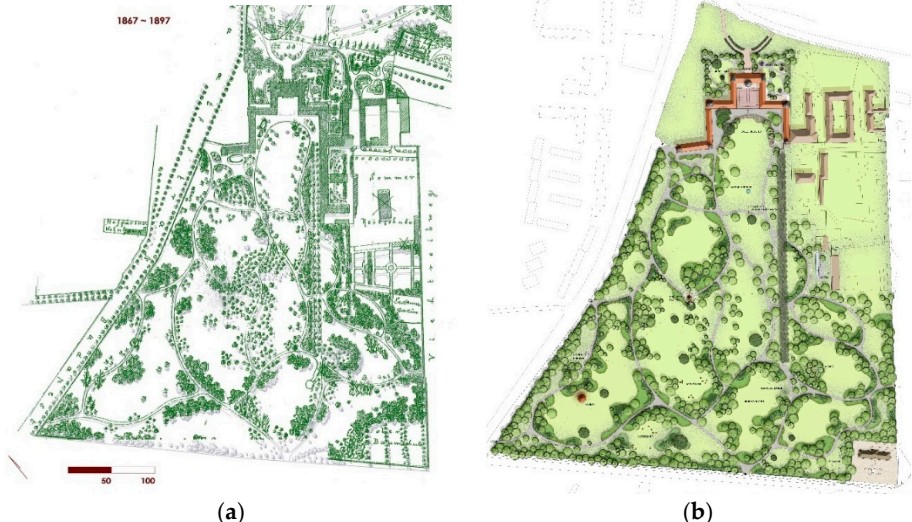

(**a**)                                                       (**b**)

**Figure 14.** The restoration of visual links is related to a successful reconstruction of the spatial layout of the garden. The spatial layout is mainly determined by the vegetation, paths, and landforms. The restored path network (**b**) entirely corresponds to the one that existed in the Romantic landscape garden (**a**) in the 2nd half of the 19th century (layouts from 1867). Design by Fekete, A.; Jámbor, I.; Sárospataki, M.; Szilágyi, K.; Vajda, Sz. 2009–2010.

The restoration of the Upper Garden in Gödöllő was based on the pictorial and written sources available. A grass surfaced pleasure ground with an approximately 300-meter long visual axis provides an appropriate vista for the impressive building of the palace. Part of the garden near the palace is, thus, visually open since this is essential for the perception of the spatial layout and scales. Regarding other parts of the garden, the plan proposed the restoration of the historic network of paths. The romantic spatial composition with groves and confined spaces here and wide open views there, which was one characterised by the most refined English style, has been created with the use of vegetated boundaries and ornamental plantings [33,34] (Figure 15b,c).

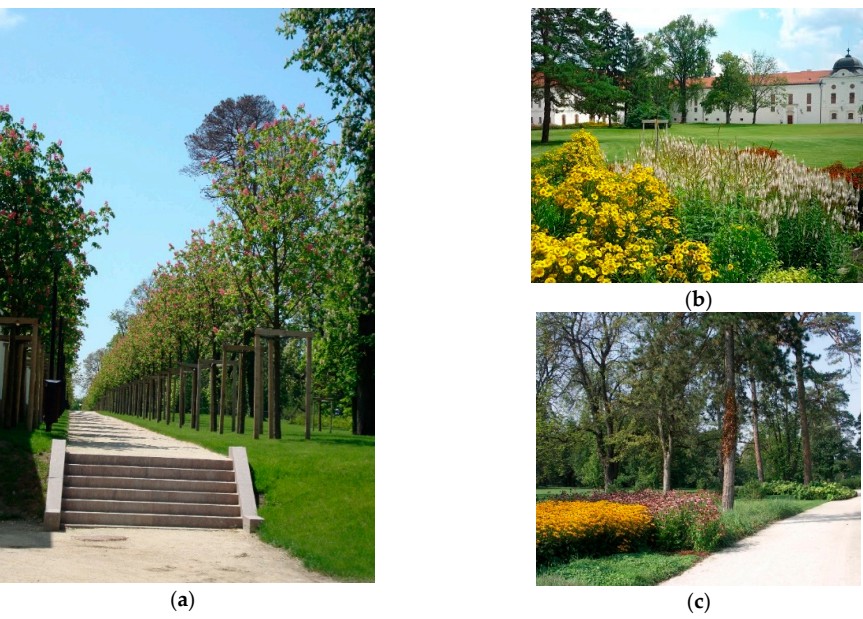

(**a**)                                                       (**c**)

**Figure 15.** (**a**) The restoration of the 400-m long horse chestnut alley, defining the baroque vista from the beginning of 18th century. 16. **b**, **c**. Colourful perennial boundaries, acting as space-defining elements and as visual focus points, too. Photos by Fekete, A, 2012 (**a**), 2016 (**b**,**c**).

Regarding internal visual links, determining a transverse visual axis, the Baroque pavilion on the Royal Mound is of outstanding importance. The full body seated bronze statue of Maria Theresa was placed on this axis as an eye-catcher.

The completely renovated, almost 400-m long horse chestnut alley of 120 trees is also of Baroque origin and determines a direction and vista parallel to the main axis, closed by the statue of Hercules with a cudgel in his hands. The alley starts from a flight of stairs made of monolith granite porphyry, flanked by short walls (Figure 15a).

## 5. Conclusions

In Hungary and especially in Transylvania, the current conditions of historic gardens and the scarce availability of historical sources on garden history in most of the cases do not allow for a full restoration of the original design. Despite this, the restoration or renovation of our historic castle gardens is an actual necessity.

Taking into consideration the driving forces (of natural or human origin) behind the change of landscapes and gardens, the original conditions of planned visual links of a garden may change during the times. A bi-directional visual link may become unidirectional or completely perish. The main factors that can have an impact on visual links may be of natural or human origin. Most of all, the loss of the former eye-catcher, the absence (disappearance, damage, relocation) of the architectural/artistic work, is the reason behind the change. Sometimes the change or the loss of a visual link may be the result of natural processes (and the lack of human intervention), for example, with the development of a forest cover. Occasionally, the modified or lost visual link is possible to restore during the landscape and garden restorations. Eye-catchers not only provide aesthetic experience but, through their positions and functions, also deliver messages. They are objects of symbolic occupation of space and represent national identity.

If scarce authentic information is at hand, contemporary tools are applied in order to evoke the atmosphere of the former garden, or the possibility of creating a new work arises.

Renovation should increase the value of the garden, ensuring the authenticity of the restoration. What is decisive during the process of restoration in the case of historic gardens is the application of the original spatial structure, spatial- and mass proportions, the restoration of the contemporary uniform composition.

During the restoration of the built elements (roads, pavements, garden structures, fixtures, etc.), the correct use of materials and traditional technologies is important. The distinction between original and restored elements, however, is expected to be unaffected even when using the same material, based on the principle of monumental authenticity. During the restoration of historic gardens, the application of the original plant species may prove difficult due to newly emerging pathogens or to the often limited variety of plants readily available. In this case, the use of substitute plant species is permitted.

The picturesqueness and the visually encoded message of the castle gardens, as well as the visual relationship between the castle estate and the surrounding landscape, are important for the conservation and restoration of our historic cultural landscapes. Owing to their visual features, artistic compositions, agelong processes of existence, and messages delivered, historic gardens bear a significant emotional substance. Their restoration is important also for experiencing natural processes. Redefinition of the relationships between the society and the heritage, humankind, and the landscape is a duty of all periods, and it urges us to always find the appropriate tools to do so.

Despite the difficulties in exploring their past, from the surveys of the actual conditions, it is possible to establish that our historic castle gardens still represent a part of our garden heritage that is possible to restore in the most authentic way. Regarding garden reconstructions, if we have no sufficient authentic information available on the original conditions, then there is the opportunity to recall the atmosphere of the former garden or occasionally to create a contemporary work of art.

The value of the presented renewals lies in the vast experience gained through the preparation of the inventor, which comprises a garden and landscape history review of nearly 400 years and complex

documentation of numerous sites. Results highlight that manor estates were complex and functional units regarding their role in arts, society, and economy. Through their establishment, the maintenance, and the various agricultural activities contributing to economic sustainability, they had significant impacts on landscapes. The planners and owners made them an important tool and element of the transformation of the 18th- and 19th-century landscapes, and they still play a significant role in the character of the landscape nowadays. Their survey and documentation has an inestimable value for heritage conservation and is absolutely necessary for any future restorations. The research project won the Europa Nostra award in the research category in 2014, and the assessment by the jury highlights that the subject is one of great importance and has revealed something of a gap in our understanding of European garden history. The outcome provides the means not only to tackle the problems of decline and dereliction in the castle gardens themselves but also to learn ways in which other neglected gardens and garden-landscape relations in other regions can be rescued and restored [35].

**Author Contributions:** Conceptualization, A.F.; methodology, A.F. and L.K.; software, L.K.; validation, A.F. and L.K.; formal analysis, A.F.; investigation, A.F.; resources, A.F. and L.K.; data curation, A.F.; writing—original draft preparation, A.F.; writing—review and editing, A.F. and L.K.; visualization, A.F. and L.K.; supervision, A.F. and L.K.; project administration, A.F.; funding acquisition, A.F., L.K.

**Acknowledgments:** This material was published using the generous support of Fábos Foundation. Any opinions, findings, and conclusions expressed in this material are those of the authors and do not necessarily reflect the views of the Foundation.

**Conflicts of Interest:** The authors declare no conflict of interest.

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
