# Peer review of "Research-Based Design Approaches in Historic Garden Renovation"

_land, doi:10.3390/land8120192_

Round 1
Reviewer 1 Report
The main topic of the research fits with the journal's scope, and it is very interesting to explain the actual case of the Hungarian Castle Garden restoration process. As research has been carried out with a lot of manpower for a long time, the completion is also high. The structure and content of the thesis are so complete that no modification is required. If correcting some grammatical errors or expressions in English, I think it will be a more complete paper.
Author Response
Response to Reviewer 1 comments
Point 1
The main topic of the research fits with the journal's scope, and it is very interesting to explain the actual case of the Hungarian Castle Garden restoration process. As research has been carried out with a lot of manpower for a long time, the completion is also high. The structure and content of the thesis are so complete that no modification is required. If correcting some grammatical errors or expressions in English, I think it will be a more complete paper.
Response 1
In spite of the fact you considere the structure and content of the paper complete and you didn’t required any modifications, we reworked it partially, accordingly to the observations received from the Reviewer 2.
A proofreading by a native english lector consider it as done.
Reviewer 2 Report
The manuscript “Research based design approaches in historic garden renovation” by Fekete and Kollányi presents a framework for evaluating historic gardens and deriving pathways for sustainable restoration.
First off, my professional background is in landscape ecology. While we do cover landscape history, including historic human perception of landscapes, and often turn to landscape architecture and landscape garden design mostly for contrasts with ecological functions, I am by no means an expert for the science covered here. That being said, I found the overall topic interesting and enjoyed reading the work. I especially liked the aspect of including current landscape context into garden “conservation” planning (I do not know if this is new in this field of research, but it is a general principle in landsape ecology). My major concern with the manuscript is, that I do not quite understand why the manuscript is forced into the structure of an experimental paper (which is why I selected 'Not applicable' in the recommendation options). Having read the whole work, the separation into introduction, methods, results and discussion makes little sense – and is also poorly adhered to by the authors (please see specific comments below). To me, the content would be much more suitable for a review-style article, separated into:
An Introduction that basically lays out the general necessity to a multi-pathway approach to reasonable historic garden restoration mainly historic research and site surveys, local garden scale and landscape scale) A “method” section that provides a clear categorization of necessary working steps for each section (e.g. family history data for historic research, relevant features at the garden scale and landscape scale). Principally, this could be provided in tables with some introductory text. Finally the examples with a clear emphasis on what principles of the framework have been at work and how they turned out. Conclusion
Generally, most of this information is already in the manuscript (and relevant, I think), I just find the arrangement extremely confusing. It just should be taken care of rigorously, that a stringent terminology is adhered to. Terms need to be introduced at the proper sections and used throughout. Now, for example, the term “prospects” comes extremely late, “restauration” (used in the title) becomes only relevant in the (now) discussion section, I was often not clear to me whether “eye-catcher are relevant on the garden or landscape scale, what exactly are “features” etc.
I am not quite sure about journal requirements, but I have seen some examples online that have a more meaningful order of content with similar studies. If put in a similar way, I think the work would make an interesting contribution (even for an only marginally targeted person like me).
Lastly, I would have a native reader check on spelling, grammar and style. I commented on some examples, but as this is not within my core expertise, I might have missed a lot.
Specific comments
L43: remove “,” after “Basin”
L50: add “,” after “history”
L51: “a” garden image?
L55: remove “.” after “elements”
L73: “.” after [12]
L79-80: There is something wrong with the grammar here. I guess the full stop at the end of line 79 should be a comma, but that is an awfully long and complicated sentence.
L86: remove “under”
L90-103: I am not sure if this is material and methods for this work or aims at describing the data available for this study. In the latter case, “something of a gap” (L100) and “outcome” (L101) is not helpful without further information (if it is methodology for this work, it needs to go into discussion/conclusion). What are the direct lines of action that were identified and are put into place in this work? Also in that case, I would put it in the introduction.
L107: “may” what? May be interpreted and therefore must be interpreted makes little sense to me.
L111-116: so does “sites” here relate to the “visual links and eye-catchers”? “Sites” to me is misleading as, in a landscape context, it could also relate to the gardens itself.
L128: Not sure what “these” relates to. I assume something like “Family history data proved to be especially important, as it was tightly linked to the initial creation or later transformation of…”
L137: “garden”
L140: I would simply say “condition of each castle garden and their valuable features” – and drop the “possible to preserve” part
L145: I don’t quite understand “The survey form records site data…”
L150-155: I am b bit surprised why an English garden appears here prominently with an own figure – I understand this is not part of this work?
L162-164: I don’t quite understand the figure. Is the redish area one example of a garden surveyed at the landscape scale, including a church (M1) and a crypt that are not part of the garden itself?
L165-167: At this point I am not sure what exactly was surveyed. I would consider a hierarchical approach starting with the number of gardens that where surveyed and what was done a) within gardens (i.e. what “features” were mapped) and then the landscape scale aspects, similar to here: “for the XY gardens surveyed, we investigated 139 build features within landscapes in order to…”
L175: “functional” in terms of what? Functional in themselves or functional for the garden design?
L177-179: Is this results already?
L189: one “garden” too many? Or “gardens” for the second one?
L181-204: I don’t see how these are results. No data is presented and the issue of “broken” gardens is mentioned here for the first time.
L206-213: How did you get to these results? Most of this is mentioned similarly in the Introduction or the method section.
L215-234: Same as above. Also, the term “prospect” appears here for the first time (except in one figure legend). If this was a study to be presented in the style of a scientific experiment (i.e. intro, methods, results, discussion), I would have expected these theses at the end of the intro, then a description of how it is tested in the methods, then the results showing the data supporting or rejecting these theses, and then a general discussion.
L236: Here for example, I would have expected the numbers of objects identified for each class of “eye-cather”
L252: Again, how have you distinguished that?
L258: There is a noticeable change in style here. The discussion seems more a separated and more independent part of the manuscript, picking up some of the aspects before, but mostly providing examples. I stopped with specific comments here, but see general comments above.
Author Response
Response to Reviewer 2 comments
Point 1:
The manuscript “Research based design approaches in historic garden renovation” by Fekete and Kollányi presents a framework for evaluating historic gardens and deriving pathways for sustainable restoration.
First off, my professional background is in landscape ecology. While we do cover landscape history, including historic human perception of landscapes, and often turn to landscape architecture and landscape garden design mostly for contrasts with ecological functions, I am by no means an expert for the science covered here. That being said, I found the overall topic interesting and enjoyed reading the work. I especially liked the aspect of including current landscape context into garden “conservation” planning (I do not know if this is new in this field of research, but it is a general principle in landsape ecology). My major concern with the manuscript is, that I do not quite understand why the manuscript is forced into the structure of an experimental paper (which is why I selected 'Not applicable' in the recommendation options). Having read the whole work, the separation into introduction, methods, results and discussion makes little sense – and is also poorly adhered to by the authors (please see specific comments below). To me, the content would be much more suitable for a review-style article, separated into:
An Introduction that basically lays out the general necessity to a multi-pathway approach to reasonable historic garden restoration mainly historic research and site surveys, local garden scale and landscape scale) A “method” section that provides a clear categorization of necessary working steps for each section (e.g. family history data for historic research, relevant features at the garden scale and landscape scale). Principally, this could be provided in tables with some introductory text. Finally the examples with a clear emphasis on what principles of the framework have been at work and how they turned out.
Generally, most of this information is already in the manuscript (and relevant, I think), I just find the arrangement extremely confusing. It just should be taken care of rigorously, that a stringent terminology is adhered to. Terms need to be introduced at the proper sections and used throughout. Now, for example, the term “prospects” comes extremely late, “restauration” (used in the title) becomes only relevant in the (now) discussion section, I was often not clear to me whether “eye-catcher are relevant on the garden or landscape scale, what exactly are “features” etc.
I am not quite sure about journal requirements, but I have seen some examples online that have a more meaningful order of content with similar studies. If put in a similar way, I think the work would make an interesting contribution (even for an only marginally targeted person like me).
Lastly, I would have a native reader check on spelling, grammar and style. I commented on some examples, but as this is not within my core expertise, I might have missed a lot.
Response 1
We partially reorganized and completed the article accordingly to general comments. The structure and the arrangement of the chapters was prepared following the guidelines of the journal, that’s why we didn’t modified it.
A proofreading by a native english lector consider it as done.
Point 2:
Specific comments
L43: remove “,” after “Basin”
L50: add “,” after “history”
L51: “a” garden image?
L55: remove “.” after “elements”
L73: “.” after [12]
L79-80: There is something wrong with the grammar here. I guess the full stop at the end of line 79 should be a comma, but that is an awfully long and complicated sentence.
L86: remove “under”
Response 2
We corrected the grammatical errors.
Point 3
L90-103: I am not sure if this is material and methods for this work or aims at describing the data available for this study. In the latter case, “something of a gap” (L100) and “outcome” (L101) is not helpful without further information (if it is methodology for this work, it needs to go into discussion/conclusion). What are the direct lines of action that were identified and are put into place in this work? Also in that case, I would put it in the introduction.
Response 3
We moved to the introduction.
Point 4
L107: “may” what? May be interpreted and therefore must be interpreted makes little sense to me.
L111-116: so does “sites” here relate to the “visual links and eye-catchers”? “Sites” to me is misleading as, in a landscape context, it could also relate to the gardens itself.
L128: Not sure what “these” relates to. I assume something like “Family history data proved to be especially important, as it was tightly linked to the initial creation or later transformation of…”
L137: “garden”
L140: I would simply say “condition of each castle garden and their valuable features” – and drop the “possible to preserve” part
L145: I don’t quite understand “The survey form records site data…”
L150-155: I am b bit surprised why an English garden appears here prominently with an own figure – I understand this is not part of this work?
Response 4
We corrected the grammatical errors, we removed the figure in charge which wasn’t very explicit, generating misunderstanding.
Point 5
L162-164: I don’t quite understand the figure. Is the redish area one example of a garden surveyed at the landscape scale, including a church (M1) and a crypt that are not part of the garden itself?
Response 5
Exactly. the church and the crypt are not part of the garden, being located outside of the borders, bu tare visually linked with the garden.
Point 6
L165-167: At this point I am not sure what exactly was surveyed. I would consider a hierarchical approach starting with the number of gardens that where surveyed and what was done a) within gardens (i.e. what “features” were mapped) and then the landscape scale aspects, similar to here: “for the XY gardens surveyed, we investigated 139 build features within landscapes in order to…”
Response 6
Reformulated,
“Out of 100 investigated locations we found in 61 cases altogether 139 built features (eye-catchers) on site, which proves the garden and landscape compositional role of eye catchers, and can serve as a basis for the renewal of the visual communication between garden and landscape. (Figure 3.)”
moved (to L 213-216) and completed with a figure (Figure 3.)
Point 7
L175: “functional” in terms of what? Functional in themselves or functional for the garden design?
Response 7
We corrected “were not really functional” with “were not really functional in themselves” (L 154)
Point 8
L181 L189: one “garden” too many? Or “gardens” for the second one?
Response 8
Corrected
Point 9
L206-213: How did you get to these results? Most of this is mentioned similarly in the Introduction or the method section.
Response 9
From the 100 investigated locations in 79 cases we discovered during the field survey a still existing strong visual connections between the castle, the garden and the landscape.
Point 10
L215-234: Same as above. Also, the term “prospect” appears here for the first time (except in one figure legend). If this was a study to be presented in the style of a scientific experiment (i.e. intro, methods, results, discussion), I would have expected these theses at the end of the intro, then a description of how it is tested in the methods, then the results showing the data supporting or rejecting these theses, and then a general discussion.
L236: Here for example, I would have expected the numbers of objects identified for each class of “eye-cather”
L252: Again, how have you distinguished that?
Response 10
We tried to complete the subchapter to be more clear and understandable.